# Whitebark Pine Recruitment in Sierra Nevada Driven by Range Position and Disturbance History

**Michèle R. Slaton [1,\*], Martin MacKenzie [2], Tanya Kohler [1] and Carlos M. Ramirez [1]**

[1] Remote Sensing Laboratory, USDA Forest Service, Pacific Southwest Region, 3237 Peacekeeper Way, Suite 201, McClellan, CA 95652, USA; tkohler@fs.fed.us (T.K.); carlosramirez@fs.fed.us (C.M.R.)

[2] Forest Health Protection, USDA Forest Service, Pacific Southwest Region, South Sierra Shared Service Area, 19777 Greenley Road, Sonora, CA 95370, USA; mmackenzie@fs.fed.us

\* Correspondence: michele.slaton@usda.gov; Tel.: +760-873-2498

**Abstract:** Effective restoration of whitebark pine populations will require a solid understanding of factors affecting seedling recruitment success, which may vary by site and biogeographic region. We examined the relationship between whitebark pine seedling recruitment, disturbance history, and range position in three independent studies in the southern Sierra Nevada, California (CA), USA. In 66 plots broadly distributed across watersheds, we found that whitebark pine seedling density and proportion were greatest at upper elevations, and where canopy cover of whitebark pine was higher (density ranged 0–383 seedlings/ha; $\bar{x} = 4$, $\sigma_X = 1$). Seedling density and proportion were also greater in plots that had recently experienced loss of canopy cover from insects, avalanche, windthrow, or other disturbance effects. In a second study conducted in popular recreational areas, including campgrounds and trailheads, the response of whitebark pine recruitment to disturbance was strongly dependent on the relative position of stands within the range, or proximity to other forest types. Both studies indicated that low to moderate levels of disturbance enhanced whitebark pine recruitment, especially at its range edge, a finding consistent with the early seral status of whitebark observed in previous studies conducted elsewhere in North America. In our third study, a case study at the June Mt. Ski Area, we demonstrate the potential for a downward shift in the whitebark-lodgepole pine ecotone as a result of insect-caused disturbance.

**Keywords:** whitebark pine; tree recruitment; population dynamics; disturbance; range distribution

## 1. Introduction

Whitebark pine (*Pinus albicaulis* Engelm.) is endemic to western North America, and in California (CA) occurs in the Sierra Nevada, Great Basin, Cascade and Klamath Mts. Found mainly above 2500 m, and often dominating the upper treeline, whitebark pine forests receive a large proportion of mountain snowfall, and therefore play an important role in regulating water supply [1]. Whitebark pine is also a candidate species for listing under the US Endangered Species Act, with primary threats from white pine blister rust (*Cronartium ribicola* J.C. Fisch.), mountain pine beetle (*Dendroctonus ponderosae* Hopkins), warmer temperatures, loss of snow, and fire exclusion [2]. For a special status species, whitebark pine is relatively common in California, occupying over 80,000 ha [3]. The fact that populations are in inaccessible terrain, and often within wilderness areas also means that the management toolset for restoration is limited, underscoring the need for highly targeted restoration efforts [4–6].

At the highest elevations where it occurs, whitebark pine is typically the sole tree species, but at lower elevations, near its range edge, other conifers may be co-dominant. In California, these species include limber pine (*P. flexilis* E. James), western white pine (*P. monticola* Douglas ex D. Don), foxtail pine (*P. balfouriana* Grev. & Balf.), lodgepole pine (*P. contorta* Loudon), red or white fir (*Abies magnifica*

A. Murray bis, *A. concolor* (Gordon & Glend.) Lindl. ex Hildebr.), mountain hemlock (*Tsuga mertensiana* (Bong.) Carrière), western juniper (*Juniperus grandis* R.P. Adams), and occasionally Jeffrey pine (*P. jeffreyi* Grev. & Balf.) (Figure 1).

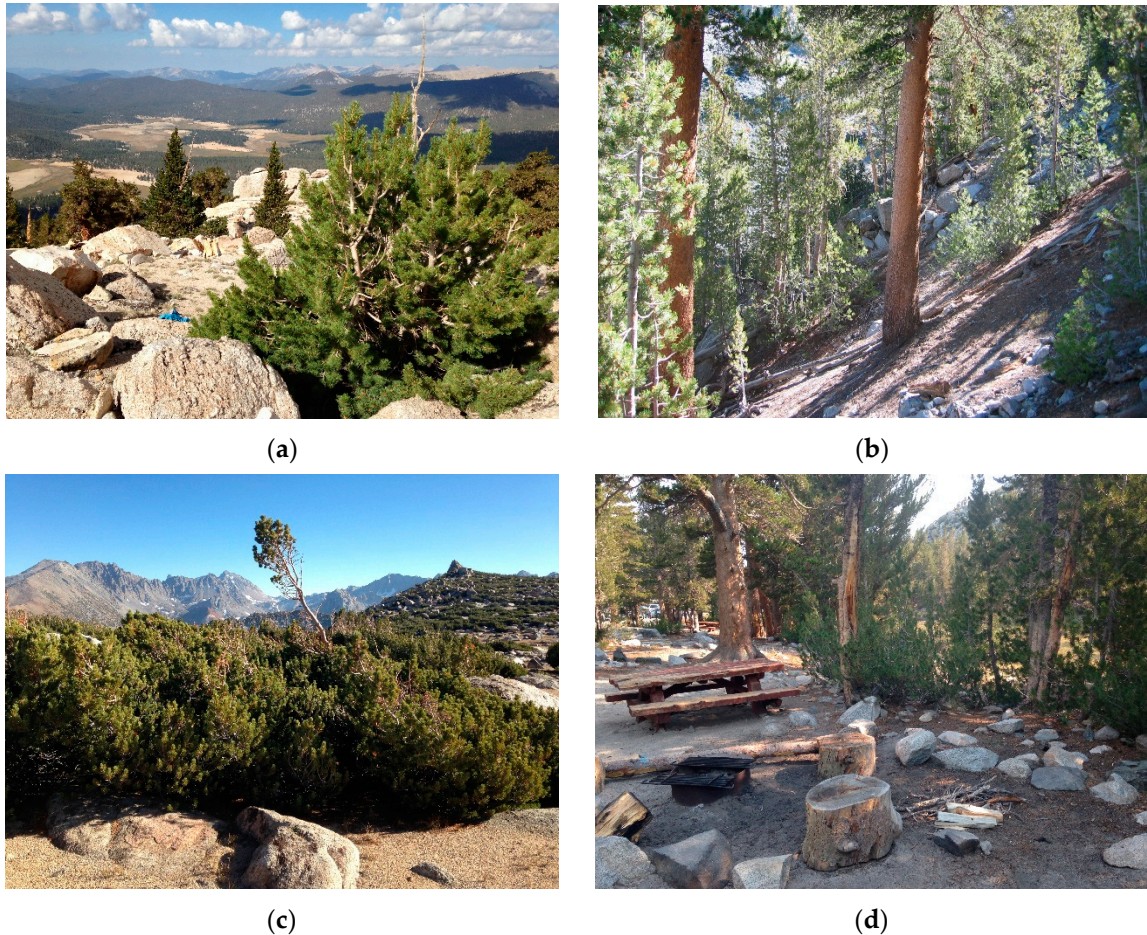

(**a**)　　　　　　　　　　　　　　　　　(**b**)

(**c**)　　　　　　　　　　　　　　　　　(**d**)

**Figure 1.** (**a**) Southern-most known whitebark pine population in North America, overlooking the Kern Plateau in southern Sierra Nevada; (**b**) study plot in the Glass Mountains, at the lower elevation range limit, where whitebark pine occurs with lodgepole pine; (**c**) study plot in Bishop Creek of the Sierra Nevada, at the upper elevation range limit, and (**d**) a campground included in the recreational study site at Rock Creek.

Whitebark pine is known to be an early successional species in the Rocky Mountains and Pacific Northwest, colonizing recently disturbed sites, including burns and mountain pine beetle attacks, and losing canopy dominance to more shade-tolerant species after 150–400 years without disturbance, especially at lower elevations [7–12]. Meyer et al. [13] observed a shift in stand structure toward smaller diameter whitebark pine stems following a mountain pine beetle attack at the June Mt. Ski Area in Mono County, CA, but patterns of subsequent succession in species composition following disturbance have not been widely documented in this state. Thus, our first objective was to examine the relationship between whitebark pine recruitment and disturbance in the southern Sierra Nevada, CA. We investigated that relationship directly, realizing that several factors—both autecological (e.g., soil temperature-related) or synecological (e.g., granivore dispersers of seeds)—likely act as underlying mechanisms. Indeed, the seeds of whitebark pine are primarily dispersed by the bird, Clark's nutcracker (*Nucifraga Columbiana* (Wilson)) [11].

Our second objective was to determine whether that relationship was dependent on the position within whitebark pine range, i.e., does proximity to other forest types affect the importance of

disturbance to whitebark pine recruitment? Populations are often analyzed as if demography were uniform throughout a species' range, although that is rarely the case [14]. Demographic studies have more recently considered range-wide variation in ecological process, where population performance metrics, such as growth and reproduction, differ by range position, i.e., between the leading and rear edges and the core [14–16]. Many authors have documented variability of whitebark pine health and regeneration across its range, as correlated to climate, elevation, tree density, tree size, or habitat [17–21]. These studies allow us to now refine potential presumptions of uniformity across the range: " . . . major threats, predation, fire and fire suppression, and environmental effects of climate change . . . also occur throughout the entire range and have resulted in significant loss of whitebark pine" [2,22].

We aimed to evaluate the abundance of whitebark pine recruitment and the relative success of whitebark compared to other tree species with respect to disturbance history, climate, and contextual range position. We hypothesized that whitebark pine recruitment would be greatest in areas recently disturbed by insects or disease, fire, avalanche, or other factors that exposed bare soil or created canopy openings. In addition, we hypothesized that the relative proportion of whitebark pine recruitment would vary by spatial position and context compared to other species, and particularly in relation to overstory tree composition. We tested these hypotheses using two independent datasets: one including 66 plots across the southern extent of whitebark pine range, and the second using intensive tree surveys at recreation sites in four separate watersheds. Inclusion of both studies here facilitated the examination of our hypotheses at two different scales and through different measures of recruitment and disturbance. Finally, we present observations from a localized site at a ski area, with a recent severe mountain pine beetle attack, as a further case study for validation. Using these three varied approaches increased our ability to interpret patterns of whitebark pine recruitment—a phenomenon which occurs in relatively low densities and with great variability across the landscape, and for which there is little information in the Sierra Nevada, which represents a major segment of whitebark pine's distribution.

## 2. Materials and Methods

### 2.1. Plot-Based Model Development

#### 2.1.1. Field Sampling Design

We sampled 66 circular plots (0.08 ha) in the southeastern Sierra Nevada and nearby Glass Mountains of the Great Basin, beginning near the southern range edge of whitebark pine in North America and extending northward. Plots were distributed across an 800 m elevational gradient and a 200 km latitudinal gradient. Plots were evenly stratified across elevation zones, but were more common in northerly latitudes, because whitebark pine distribution is highly restricted at its southern range edge (Figure 2). Whitebark forests here remain relatively healthy as of 2019; white pine blister rust has not yet been documented in the southeastern Sierra Nevada, and mountain pine beetle attacks are relatively limited in extent [3].

Plots were sampled once during 2011–2018. Although the multi-year time span of data collection limits our ability to make assertions regarding the relation of whitebark pine recruitment to annual variability in precipitation or temperature, it does not preclude hypothesis-testing regarding recruitment's relation to range position and average climate conditions. Whitebark pine regeneration is known to be episodic, through masting, but we saw no major surges or trends through time in germination that would skew interpretation, and disturbance categories and seedling age classes were defined broadly enough to reduce the importance of temporal specificity.

In 31 plots, whitebark pine was the sole tree species; the remaining 35 plots contained at least one other tree species. Within the stratified zones, general locations were selected for accessibility, only requiring that whitebark be present in the plot, and that slope and aspect be relatively uniform across the plot. Plot centers were selected randomly by pacing at least 15 m from access trails or other entry points into whitebark pine habitat.

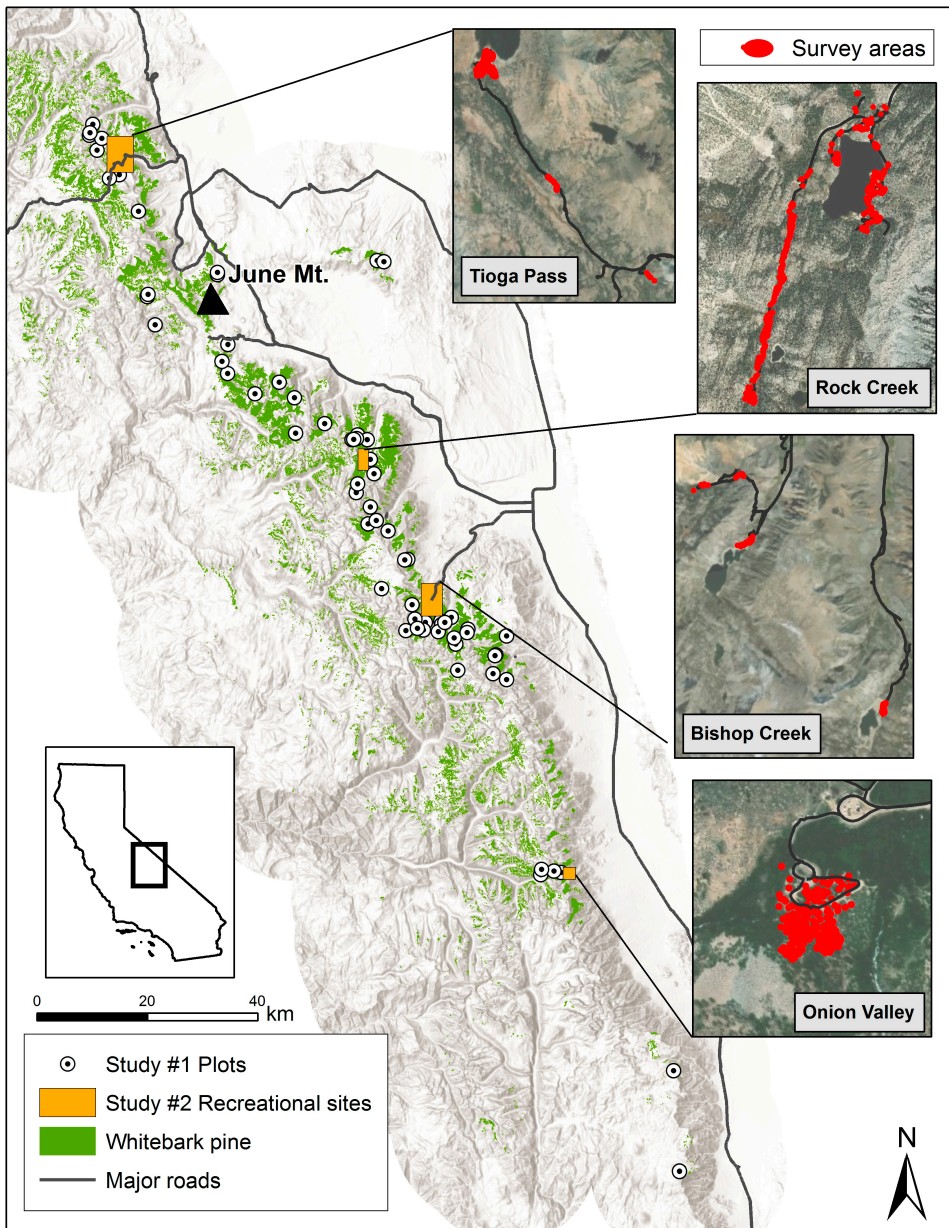

**Figure 2.** Map of study area in eastern California, USA, displaying Study #1 plots, the four recreational sites surveyed in Study #2, and the June Mt. case study area.

In each plot, we counted the number of trees of each species that were less than 1.37 m in height. Tree height is not necessarily a good indicator of age, especially in whitebark pine, which can grow near or along the ground surface, where it remains protected under snow during the winter months. To account for this, we estimated seedling age to the nearest 5 years based on terminal bud scars and whorl and branching patterns. As shown in greater detail in the June Mt. case study, it was typical for seedlings under 0.5 m in height to be up to 55 years old. Seedlings were thus defined as trees <1.37 m height and with an estimated age less than 30 years. Individuals in clusters were counted separately unless there was clear evidence of attachment to another stem. Basal sprouts or layered branches attached to older stems were not included. A narrower definition for seedlings was considered (e.g., <10 years, to better match our disturbance categories as described below). However, because so few young seedlings were present on the landscape, we elected the broader definition to be more inclusive, acknowledging that precise ages for individuals 10–30 years old is difficult without tree ring analysis.

Canopy cover was measured for each plot in the field by estimating the vertical projection of tree crown perimeters as if they were viewed from above. Canopy cover was recorded for each tree species by strata, categorized by stem length: regeneration layer (<1.37 m), midstory (1.37–4 m), and overstory (>4 m). Within each plot, we recorded every herb and shrub species, with total canopy cover recorded for each, height of shrubs (average of three nearest plot center), and approximate age categories for shrubs, to assist in ranking of disturbance history, as described below. Shrub age categories included (1) seedling, with cotyledons still evident, or a single mature leaf whorl present; (2) juveniles, estimated as under 10 years of age; and (3) adults, estimated as over 10 years of age. The latter two categories were differentiated based on leaf scars and branching patterns. Herbs were assessed only qualitatively by noting presence of cotyledons, the extent of clonal growth of graminoids and, evidence of previous years' growth on perennial forbs.

Disturbance history was assessed during the on-site inspection. Sites were assigned to one of six categories, ranked by severity and recency of disturbance throughout the plot: (1) no evidence of disturbance from fire, landslide, avalanche, soil erosion or rilling, or canopy cover loss caused by insects, disease, drought, or windthrow; little or no recent recruitment of perennial herbs or shrubs, i.e., late succession. (2) Evidence of disturbance within the plot, but generally >10 years old and affecting a small portion of the plot. (3) Evidence of disturbance apparent, but may be >10 years old and/or not severe, with most of the tree canopy unaffected. (4) Evidence of disturbance affecting at least half the plot, generally with woody canopy cover loss in the last 10 years, and some shrub or herbaceous recruitment. (5) Recent evidence of disturbance throughout most of the plot, with loss of live canopy cover of mature woody species. (6) Recent evidence of disturbance, with loss of most live woody and herbaceous canopy cover, and/or deposition of debris, soil, or gravel from landslide, avalanche, or debris flow, i.e., typically early stages of succession.

### 2.1.2. Model Development

To test our hypotheses regarding whitebark pine recruitment, we used two response variables in a multiple regression analysis: (1) total number of whitebark pine seedlings in the plot, and (2) proportion of the number of total tree seedlings that were whitebark. Analyses were performed in the R Studio environment [23].

We used three different predictors as proxy indicators of each plot's contextual position within whitebark pine range: latitude, elevation, and total and relative overstory canopy cover of whitebark pine. Our study area included the southern-most limit of whitebark pine in North America, and extended approximately 200 km northward. In our study area, whitebark becomes increasingly common above 2500 m, and declines in abundance at the highest elevations (>3800 m), where climate, extreme slopes, and lack of soil constrain its establishment. Field observations and inspection of mapping products in CA indicated that whitebark pine is typically the dominant species at the core (center) of its range, and becomes increasingly co-dominant with other species at its range edge; the range edge is thus defined more in terms of dominance loss than a precise geographic boundary. Topographic predictors were also assessed, including slope and aspect. Aspect was measured in the field, and then transformed to southwestness, calculated as cos(aspect −225 deg). Southwestness serves as an indicator of site heat load, with values ranging from −1 to +1, where −1 represents northeast, and +1 represents southwest.

We examined model covariance with climate predictors, including mean winter minimum temperature, mean summer maximum temperature, precipitation, snowfall, potential evapotranspiration, and climatic water deficit. Climate is known to be an important factor in whitebark pine distribution; though the role of climate was not our central focus, we explored these correlations to best specify and interpret our models, knowing that range position and climate are closely related. Climate data were obtained from the California Basin Characterization Model [24], and include 30-year annual means for total precipitation and snow (1980–2010), and mean minimum temperature for winter months (Dec.–Feb.), and mean maximum temperature for summer months (June–Aug.).

For our questions related to the importance of range position at a broad scale, elevation and latitude were used as general proxies for climate zones, and, in the case of elevation, of proximity to other forest types. Climate variables would be better addressed through studies specifically designed to isolate the influences of temperature vs precipitation, including microsite snow depth, night-sky exposure, and cold-air pooling. As a result, and because of the multicollinearity among climate variables and elevation, we eliminated climate variables as predictors from our main model (Figure 3). Instead, elevation and latitude, plus overstory species composition, served as proxies for contextual position within the distribution range of whitebark pine.

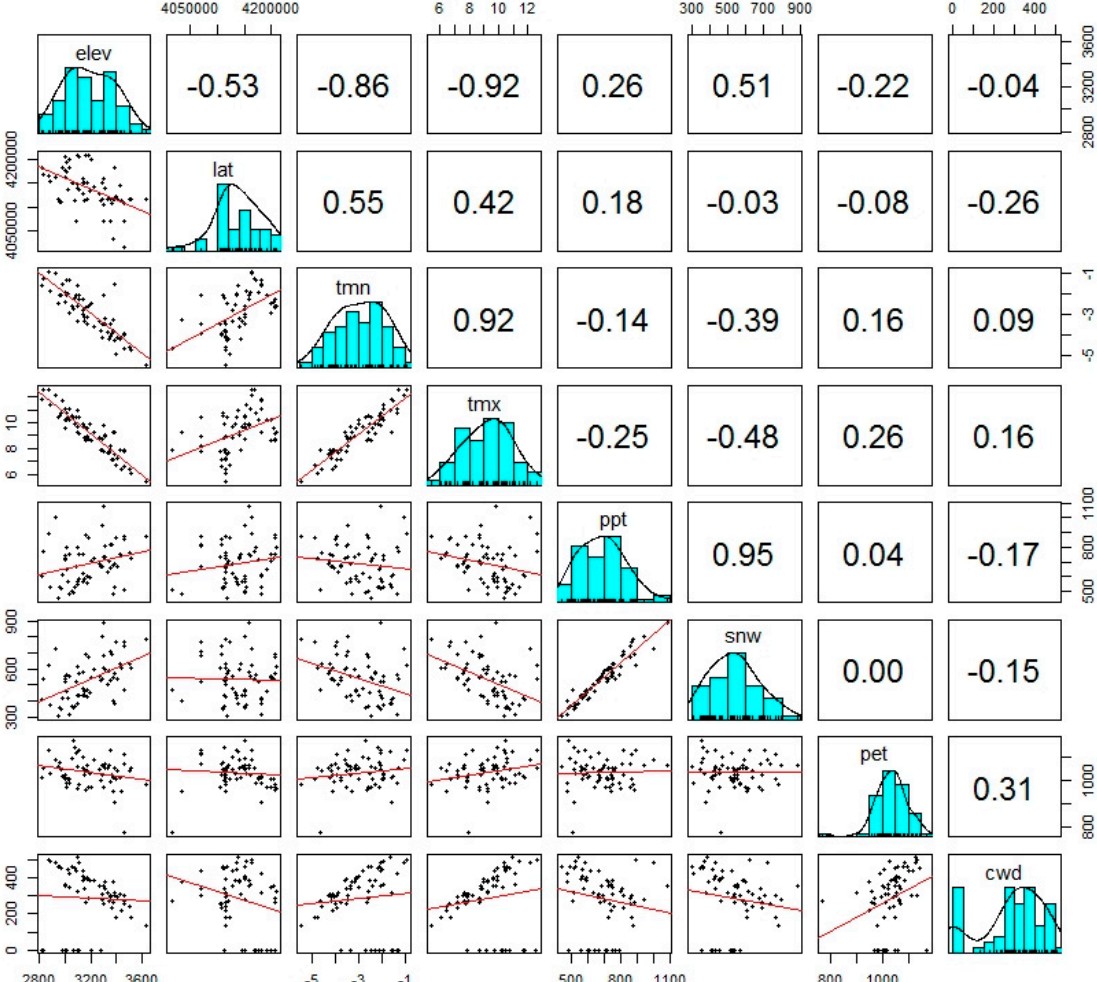

**Figure 3.** Scatterplot matrix of climate variables, including variable distributions, simple linear fits, and Pearson coefficients. elev = elevation and lat = latitude (m); tmn = mean minimum temperature and tmx = mean maximum temperature (°C); ppt = total annual precipitation, snw = total annual snowfall, pet = potential evapotranspiration, and cwd = climatic water deficit (all in units of cm).

2.1.3. Model Evaluation

We tested the assumptions of multiple linear regression through examination of the scatterplot matrix among variables, and with statistical tests where appropriate. Model residuals were examined for normality, and tested with the Anderson–Darling test. We tested for heteroscedasticity using the Breusch Pagan test, which evaluates the variance of residuals across fitted model values. Diagnostic tests were performed using *R* packages lmtest [25] and nortest [26]. Where indicated, we used standard testing procedures, including the Box Cox method to identify non-linear transformations to improve model fits. Our study aimed to test hypotheses of significant correlations among variables, and not to

predict or infer from the regression model. We therefore used departure from assumptions mainly as a tool to better understand relationships, and relied on corrective tools, such as inclusion of non-linear terms, based on model diagnostics only where they resulted in potential change to interpretation.

*2.2. Stem Density in Recreational Sites*

We used an additional independent dataset to test our hypotheses that recruitment of whitebark pine is correlated to disturbance status and range positioning. In 2015, we conducted tree inventories at four localities in the eastern Sierra Nevada, and covering a range of disturbance severities, primarily related to infrastructure maintenance and recreational activities, including camping, hiking, and fishing. The sites included Onion Valley, Bishop Creek, Rock Creek, and Tioga Pass, spanning 2800–3110 m in elevation, and each separated in latitude by at least 40 km (Figures 1d and 2). Seedlings were defined as trees with a diameter at breast height (dbh) < 2.5 cm, and with no apparent connection to another stem (i.e., basal sprouts excluded). The seedling definition in this study is thus more inclusive than that used in Study #1. Within the study sites, all areas with whitebark pine present in the overstory were surveyed, totaling approximately 5 ha per site. Every seedling within 10 m of roads and trails was assessed, resulting in a total sample of 1420 stems, with data recorded for species, gps position, and health and disturbance effects. Other conifers in the study sites included lodgepole pine, limber pine, foxtail pine, red fir, and Jeffrey pine.

Range position for each tree was scored categorically: edge as <3 km to the lowest distribution of whitebark pine that we observed in the respective watershed, and core as >3 km from the lowest edge. Because all sites occurred within a relatively narrow elevation band, this metric for range positioning is unlikely to be highly correlated to climate; rather, we interpreted it as an indicator of proximity to seed sources from potentially competing conifers. Because our study sites were generally in the lower elevation band where whitebark pines occur, distance to the upper range edge was not considered. Disturbance severity was also scored categorically as severe, moderate, or low. Severe represented compacted soils due to trails, campsites, or recreation facilities, with minimal duff and litter, or soil disturbance due to road or trail cuts within 1 m of the dripline of the tree crown, and typically with tree stems cut nearby, thus opening the canopy; moderate represented some evidence of soil trampling, compaction, or other disturbance within 1–2 m of the dripline; low represented minimal evidence of disturbance 2–4 m radius from the dripline. Because the entire study was conducted in high-use recreational areas, no completely undisturbed areas were included. Only a single tree (not a seedling) in the study area died as a result of mountain pine beetle attack, and there was no evidence of white pine blister rust or recent fire.

Seedling density was calculated as the number of seedlings divided by the total area surveyed per analysis category (disturbance and range position), as recorded in the field using a gps unit. The design was unbalanced in that Bishop Creek and Onion Valley did not have the opportunity for core range positioning. This resulted in an effective sample of 18 (3 disturbance levels, 2 range positions at 2 sites, and one range position at 2 sites). The proportion of whitebark pine seedlings was calculated as the number of whitebark divided by the total number of seedlings of all tree species.

We used a generalized linear mixed model (GLMM) in the *R* package lme4 [27] to analyze the relationships between whitebark pine recruitment, disturbance, and range position. GLMMs provide a flexible approach for datasets with factorial designs that do not necessarily meet the assumptions for traditional ANOVA, and Bolker et al. [28] give a helpful review of their use. Our samples all occurred in one of four localized sites, resulting in the non-independence of samples; however, the GLMM approach allowed us to evaluate the relative importance of site variance compared to predictors of interest. We used disturbance and range position as fixed effects, and site (Onion Valley, Bishop Creek, Rock Creek, and Tioga Pass) as a random effect, with an unstructured variance-covariance structure. Seedling counts were fitted using a Poisson distribution, and an offset of the logarithm of the survey area to adjust for density. For % whitebark pine seedlings, we used a Gaussian distribution. For fixed effects, *p*-values based on Wald's tests are reported, with multiple comparisons made with Tukey's test

using the Bonferroni adjustment for multiple comparisons in the *R* package emmeans [29]. In addition, the significance of each predictor and of each interaction was evaluated through log-likelihood ratio tests, using chi square ($\chi^2$) as a test statistic, comparing a full model including all predictors and an interaction term, to reduced models excluding each of these. The Akaike Information Criterion (AIC) was also used to assess which predictors significantly improved the models.

### 2.3. Observational Case Study in Mountain Pine Beetle Attack at June Mt.

The June Mt. Ski Area in central Mono County was established in 1958, and ranges 2400–3000 m in elevation (Figure 2). A mountain pine beetle attack became evident in the June Mt. area in 2005. Although lodgepole pine is often listed as the primary host of the mountain pine beetle [30], the June Mt. outbreak began in whitebark pine, and proceeded downward in elevation into the secondary host, lodgepole pine, as the primary host became exhausted (Figure 4). Furniss and Renkin [31] have similarly suggested that mountain pine beetle attacks do not necessarily originate in lodgepole pine before moving into whitebark.

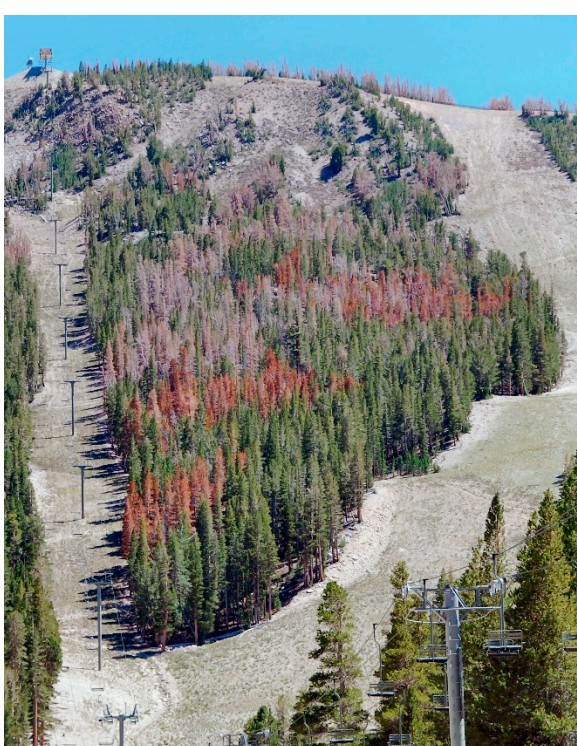

**Figure 4.** The June Mt. Ski Area case study site. Red-needled lodgepole pines were attacked by mountain pine beetles following the initial attack on whitebark pine at higher elevations.

US Forest Service Forest Health Protection staff performed semi-annual on-site surveys 2006–2018 to assess the health and recovery trajectory of the whitebark pine ecosystem and inform decisions regarding the need for active management to achieve ecosystem recovery. Surveys were mostly qualitative in nature, with emphasis on the composition and health of conifer recruitment. Several transects were surveyed across the ski runs at intermittent elevations in 2012, with adjacent areas in the intact forest also surveyed. Several seedling clusters were harvested in 2012 for age analysis. These stems were sectioned at ground level and stained. Height data were collected in 2018 from whitebark pine seedlings within the severe beetle attack area 2600–2700 m elevation on the mountain. Heights as of 2015 were obtained from the location of needle scars.

## 3. Results

### 3.1. Plot-Based Model Development

Whitebark pine recruitment was significantly correlated to the disturbance and range position, though relationships varied by metric and were in some cases weak (Figure 5, Table 1). Little covariance was found among explanatory variables. A negative correlation was evident between elevation and latitude, resulting from the geomorphology of the study area. The Sierra Nevada reach their highest elevations at the southern end of the range (at Mt. Whitney, near the southern extent of whitebark pine's range), with terrain dropping precipitously in elevation from there, and resulting in limited availability of land at lower elevations where whitebark pine might occur. In contrast, the northern end of the study area has abundant terrain and is less steep—in the 2700–3400 m elevation zone. Furthermore, the greater amount of precipitation in the north provides greater opportunity for whitebark pine to occupy lower elevation sites that otherwise are dominated by more drought-tolerant conifers (limber pine, foxtail pine) in the south. Despite the collinearity between elevation and latitude, we elected to keep both predictors in our models. The correlation was not extremely strong, and both variables provided insights for interpretation.

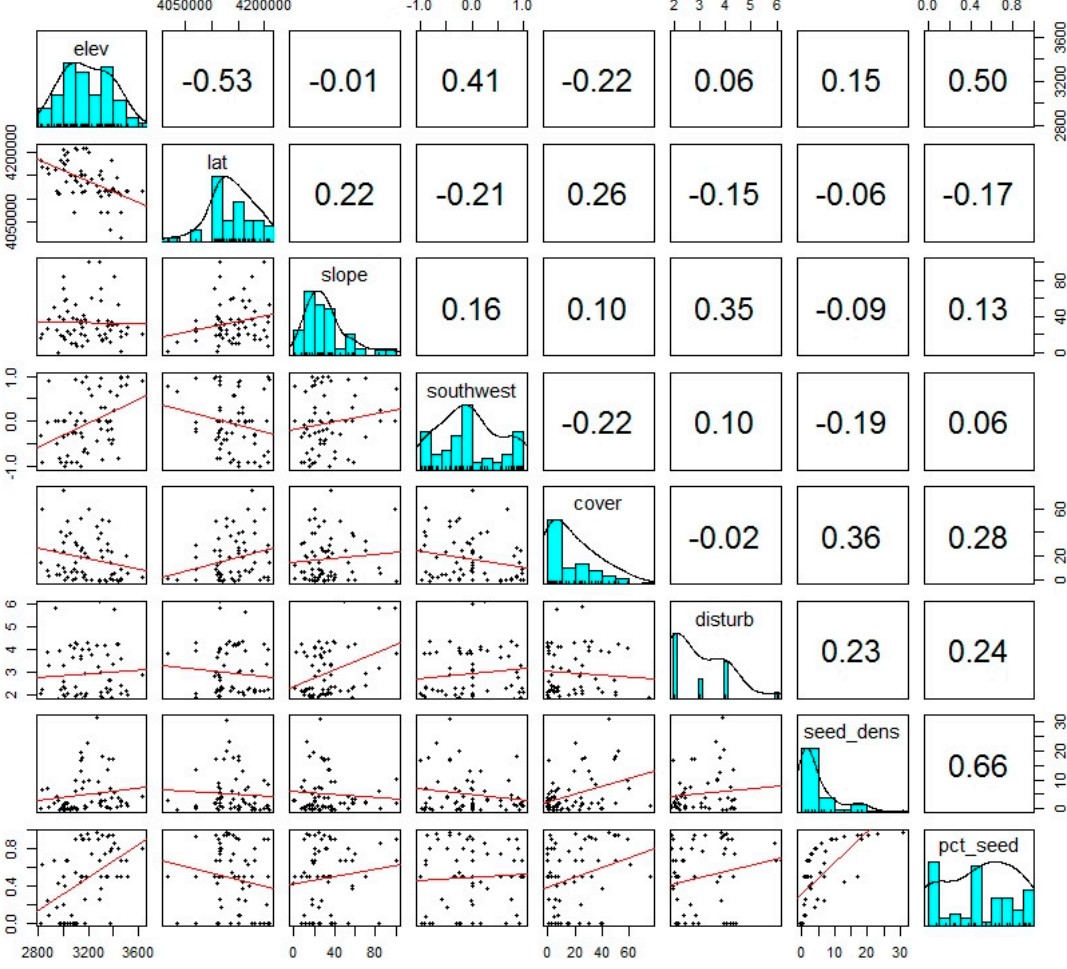

**Figure 5.** Scatterplot matrix for explanatory and response variables. elev = elevation, lat = latitude (m); southwest = southwestness, a measure of aspect; cover = main canopy cover of whitebark pine per 0.08 ha plot; disturb = categorical level of disturbance, where higher values represent greater severity and recency of disturbance; seed_dens = density of seedlings per ha; pct_seed = proportion of whitebark seedlings with respect to all conifer seedlings.

**Table 1.** Multiple regression results for Study #1, plot-based models. Explanatory variables in the first column, and response variables in the first row for each of two models. Quadratic terms for elevation and latitude were introduced into models where they improved fits.

| | Whitebark Seedling Density | Whitebark Seedling Proportion (%) |
|---|---|---|
| Disturbance severity | + ** | + * |
| Whitebark main canopy cover | + *** | + *** |
| Elevation | + * | + *** |
| Elevation^2 | * | − |
| Latitude | + * | ns |
| Latitude^2 | * | − |
| Slope | − * | ns |
| Aspect | ns | ns |
| *R*-squared | 0.32 | 0.47 |
| Breusch-Pagan Test | 0.009 | 0.08 |
| Anderson Darling Test | 0.20 | 0.71 |

Significance levels: * $\alpha = 0.1$, ** $\alpha = 0.01$, and *** $\alpha = 0.0001$. Signs (+/−) indicate direction of relationship.

Whitebark pine seedling density was positively correlated to disturbance and to the main canopy cover of whitebark (Figure 5, Table 1). However, cover in the main canopy of species other than whitebark did not significantly improve models for any response; nor did total plot cover of shrubs or herbs ($p > 0.1$). Whitebark pine seedling density was significantly, though non-linearly, correlated to elevation and to latitude, with numbers peaking at mid elevations and latitudes (Figures 5 and 6a). Both of those variables are highly correlated to climate, although climate variables provided only marginal improvements to fits, possibly as a result of our limited sample size in a large study area. Because explanatory climate variables were not our area of focus, we considered elevation and latitude as proxies for range position, recognizing range position represents the combined influence of climate plus adjacency to and relative dominance of other tree species.

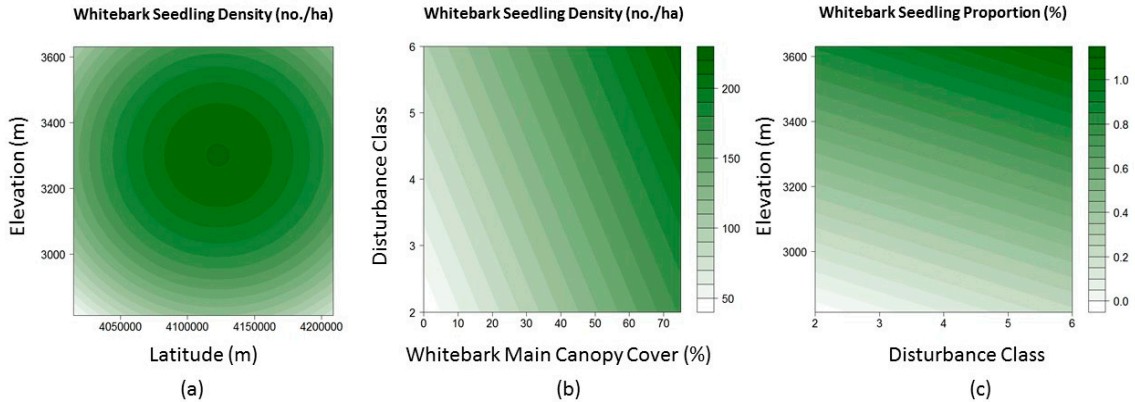

**Figure 6.** Filled contour plots displaying combined effects of explanatory variables in Study #1. Response variables shown at the top for (**a**) and (**b**) whitebark pine seedling density; (**c**) whitebark pine seedling proportion (%).

The proportion of whitebark pine seedlings was positively correlated to the same predictors as seedling density, with the exception of latitude and slope; proportion of whitebark seedlings was relatively even across latitude. Failure to detect a pattern may have been due to the small sample size and high variance. Alternatively, relatively high proportions of whitebark, despite low absolute densities, at the latitudinal edges of our study area may indicate greater tolerance, as compared to other conifers, to extreme climatic conditions, such as deep snow in the north, and aridity in the south. This pattern mirrors whitebark pine's ability to tolerate high elevations, where abundant snow and a

short growing season are coupled with high evaporative demand, warm soil temperatures, and limited water availability in the summer [32].

Some models exhibited non-normality of residuals and/or heteroscedasticity, and these issues were in part corrected by introducing non-linear functions for elevation and latitude. Even so, violations of model assumptions limit model transferability. However, they are not severe enough to change our conclusions for hypothesis tests of correlation. Unequal variances across fitted model values were potentially due to missing predictors, or to the small sample of plots with very high seedling densities. However, inclusion of additional climatic variables and transformations provided only marginal improvements to model fits.

Combined effects of key predictors are displayed in Figure 6. The greatest densities of whitebark pine seedlings were found in areas that were disturbed, but still retained high levels of canopy cover in the main stratum. The highest whitebark pine densities were also found in the most disturbed areas at higher elevations.

### 3.2. Stem Density in Recreational Sites

Parameters for the GLMM models tested are given in Table 2. Variance of random effects (site) was relatively high, in comparison to the standard error of the model intercepts. This indicates that site is an important predictor for seedling recruitment, corroborating our finding from Study #1 that broad-scale range context (e.g., elevation, latitude) is an important predictor.

**Table 2.** Whitebark pine seedling density results for Study #2 generalized linear mixed models (GLMM). Explanatory variables (disturbance and range position) are factors. Those with multiple levels are assigned codes representing predicted values from each baseline category, where intercepts represent the first category. Estimated coefficients, standard errors (SE), and *p*-values for Wald tests are shown for each variable. Akaike Information Criteria (AIC) are given, plus log-likelihood ratio test results comparing Models A, B, and D to Model C to isolate the significance of each variable, using chi square ($\chi^2$) as a test statistic. Disturb = disturbance level; range = range position; area = survey area size; and site = study locality.

| Factor | Coefficient | SE | *p*(Wald) | Variance of Random Effects | AIC | $p(\chi^2)$ Comparison to Model C |
|---|---|---|---|---|---|---|
| **Model A: seedling.density ~ range + offset(log(area)) + (1 \| site)** | | | | | | |
| | | | | 1.116 | 239.2 | 0.001 |
| Intercept | 3.94477 | 0.5435 | <0.00001 | | | |
| range.edge | −0.62307 | 0.09623 | <0.00001 | | | |
| **Model B: seedling.density ~ disturb + offset(log(area)) + (1 \| site)** | | | | | | |
| | | | | 1.592 | 266.4 | <0.00001 |
| Intercept | 3.57984 | 0.64374 | <0.00001 | | | |
| disturb.mod | −0.26221 | 0.09626 | 0.00645 | | | |
| disturb.severe | 0.12024 | 0.0976 | 0.21979 | | | |
| **Model C: seedling.density ~ disturb + range + offset(log(area)) + (1 \| site)** | | | | | | |
| | | | | 1.156 | 229.8 | na |
| Intercept | 3.98644 | 0.55606 | <0.00001 | | | |
| range.edge | −0.59628 | 0.09639 | <0.00001 | | | |
| disturb.mod | −0.25254 | 0.09774 | 0.0088 | | | |
| disturb.severe | 0.07745 | 0.09654 | 0.4281 | | | |
| **Model D: seedling.density ~ disturb + range + (disturb × range) + offset(log(area)) + (1 \| site)** | | | | | | |
| | | | | 0.9956 | 158.2 | <0.00001 |
| Intercept | 3.6453 | 0.5242 | <0.00001 | | | |
| range.edge | 0.2514 | 0.1451 | 0.0831 | | | |
| disturb.mod | 0.2964 | 0.1367 | 0.0301 | | | |
| disturb.severe | 0.7172 | 0.1321 | <0.00001 | | | |
| disturb.mod:range.edge | −1.1884 | 0.2038 | <0.00001 | | | |
| disturb.severe:range.edge | −1.9335 | 0.2545 | <0.00001 | | | |

Fixed effects were significant, and became especially strong when considering their interactions. The coefficients for fixed effects may be interpreted similarly to ordinary least squares coefficients; they give an estimate for effect size. For example, in Model A, edge localities have on average 62 fewer whitebark pine seedlings per 100 ha as compared to core localities ($p < 0.00001$; Table 2). The models with multiple fixed effects, or with effects that had multiple categories (e.g., low, moderate, high disturbance) produce unique coefficients that represent model slopes for each category as compared to a fixed baseline. These results are more easily evaluated using the AIC for model comparison (Tables 2 and 3), and visualized in Figure 7. The effect of disturbance alone, across edge and core localities is estimated by comparing Models A and C. Model C for whitebark pine seedling density does produce a lower AIC, and $p(\chi^2)$ comparing the models is 0.001; the effect of disturbance is significant, but not extremely large.

**Table 3.** Whitebark pine seedling proportion (%) results for Study #2 generalized linear mixed models (GLMM). Explanatory variables (disturbance and range position) are factors. Those with multiple levels are assigned codes representing predicted values from each baseline category, where intercepts represent the first category. Estimated coefficients, standard errors (SE), and *p*-values for Wald tests are shown for each variable. Akaike Information Criteria (AIC) are given, plus log-likelihood ratio test results comparing Models A, B, and D to Model C to isolate the significance of each variable. Disturb = disturbance level; range = range position; and site = study locality.

| Factor | Coefficient | SE | *p* (Wald) | Variance of Random Effects | AIC | $p(\chi^2)$ Comparison to Model C |
|---|---|---|---|---|---|---|
| **Model A: % seedlings ~ range + (1 \| site)** | | | | | | |
| | | | | 0.03241 | 23.9 | 0.0042 |
| Intercept | 0.479 | 0.1724 | 0.0277 | | | |
| range.edge | −0.1611 | 0.1741 | 0.3689 | | | |
| **Model B: % seedlings ~ disturb + (1 \| site)** | | | | | | |
| | | | | 0.05837 | 18.9 | 0.3032 |
| Intercept | 0.6339 | 0.1537 | 0.00845 | | | |
| disturb.mod | −0.4497 | 0.1322 | 0.00522 | | | |
| disturb.severe | −0.4097 | 0.1322 | 0.00916 | | | |
| **Model C: % seedlings ~ disturb + range + (1 \| site)** | | | | | | |
| | | | | 0.04517 | 22.4 | na |
| Intercept | 0.7265 | 0.1736 | 0.00268 | | | |
| range.edge | −0.1221 | 0.1315 | 0.3709 | | | |
| disturb.mod | −0.4497 | 0.1366 | 0.0072 | | | |
| disturb.severe | −0.4097 | 0.1366 | 0.01213 | | | |
| **Model D: % seedlings ~ disturb + range + (disturb × range) + (1 \| site)** | | | | | | |
| | | | | 0.0507 | 22 | 0.01197 |
| Intercept | 0.53301 | 0.18226 | 0.016 | | | |
| range.edge | 0.15943 | 0.17205 | 0.3775 | | | |
| disturb.mod | −0.34372 | −0.34372 | 0.1044 | | | |
| disturb.severe | 0.01277 | 0.01277 | 0.948 | | | |
| disturb.mod:range.edge | −0.15894 | 0.23308 | 0.5125 | | | |
| disturb.severe:range.edge | −0.63366 | 0.23308 | 0.0237 | | | |

The effect of range positioning was in comparison stronger; a comparison of Models B and C produces a relatively greater difference in AIC, and the likelihood ratio test comparing the models indicated significance ($p(\chi^2) < 0.0001$). Adding the interaction term (disturbance * range position) produced the best model (AIC = 158.2, $p(\chi^2) < 0.0001$). In other words, when using disturbance as the sole predictor in a model of whitebark pine seedling density, only a weak relationship was found. However, by introducing potential interaction with range positioning, disturbance had a strong effect. In fact, the effect of disturbance reversed in directionality depending on range position. Within the range core, increasing disturbance severity resulted in greater whitebark pine seedling density,

which might be expected, because seed sources are abundant in close proximity to colonize canopy openings and disturbed soils. However, at the range edge, very high disturbance magnitude reduced whitebark pine seedling density.

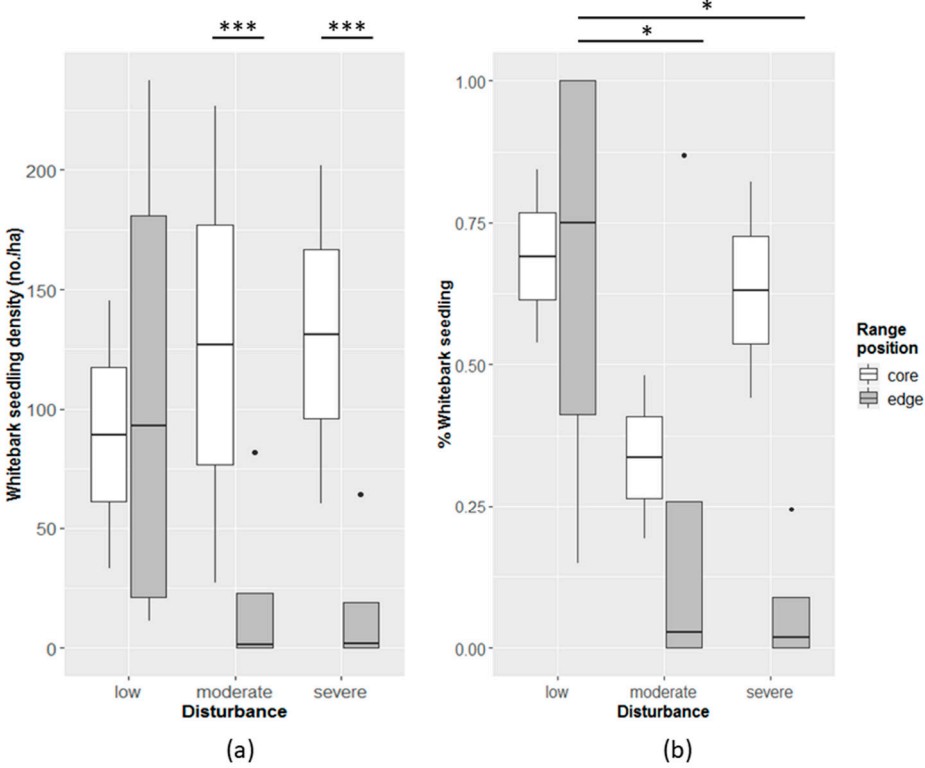

**Figure 7.** Box and whisker plots illustrating Study #2 GLMM results for (**a**) whitebark pine seedling density; and (**b**) % whitebark pine seedlings of total seedling numbers. Multiple comparisons between groups using Tukey's method are shown only for comparisons of primary interest (*** $\alpha = 0.001$ and * $\alpha = 0.1$).

The proportion of whitebark pine seedlings was also marginally affected by the interaction between disturbance and range position (Figure 6c, Table 3). In core sites, whitebark seedling proportions remained high, regardless of disturbance level. However, at the range edge, moderate to severe disturbance resulted in a strong proportional loss of whitebark pine. At most sites, with the exception of Onion Valley, lodgepole pine was the primary competitor in highly disturbed areas at the range edge. Along roadsides and trails, especially where water accumulated, lodgepole pine germinated in abundance, at the exclusion of whitebark pine. The best model for whitebark pine seedling proportion included only disturbance as an explanatory variable (AIC = 18.9, $p(\chi^2) = 0.004$ comparing Models A and C).

### 3.3. Observational Case Study in Mountain Pine Beetle Attack at June Mt.

The whitebark pine seedlings in the surveyed area adjacent to the ski runs maintained a slow growth rate prior to the beetle attack. Stems 0.6–1.2 m in height were estimated to be 55–75 years old, and a 0.2 m tall cluster of 5 trees was estimated to be 21 +/− 3 years old. In an extreme case, a seedling was observed in which the annual growth amounted to 4 to 6 tracheids per year, equating to 100 rings/cm.

The sections cut in the fall of 2012 indicated the seedlings had responded to the release caused by the overstory mortality, even when the seedlings were decades old (Figure 8). This is similar to that which has been observed in seedlings following prescribed fire and thinning treatments [33]. By 2012, needles had dropped from the beetle attacked trees; the growth ring in this year was as wide as the

three previous years combined. On average, the tallest seedlings per cluster had almost doubled their height 2015–2018 (change in height $\bar{x}$ = 196%; $\sigma_X$ 7%; *n* = 20). No whitebark pine seedlings were found in transects on the ski runs, although lodgepole pine seedlings were common. Whitebark seedlings were common on the open ridge at the top of the ski runs, and in the beetle attacked forest between the ski runs. Extensive searching in both 2012 and 2018 failed to locate any clusters of seedlings that had germinated in the previous 5 years. In all years of searching, only two clusters of seeds in the cotyledon stage were ever found. It appeared that in the decade following the initiation of the drought, which may have also initiated the bark beetle outbreak, there had never been enough moisture or abandoned seed caches to spur significant seed germination.

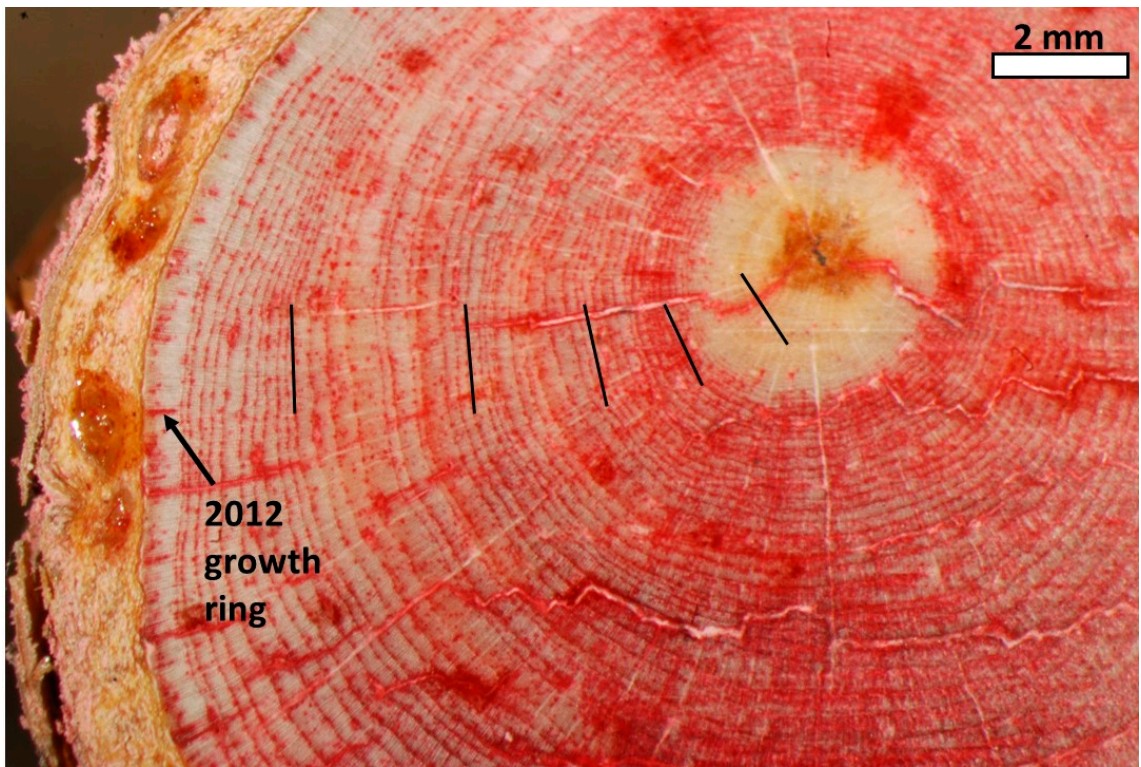

**Figure 8.** Cross-section of a whitebark pine seedling, with an estimated age of 55 years. Black lines represent 10-year intervals from the outer ring. The 2012 growth ring represents the release response following bark beetle-caused mortality of the understory.

## 4. Discussion

Each of our three studies examined hypotheses about the relationships between whitebark pine recruitment, disturbance history, and contextual range position. The use of a variety of approaches and scales provided improved understanding of the ecology of these relationships, especially given the challenge of adequately sampling a sparse and highly patchy phenomenon across a large and inaccessible landscape. We begin by discussing our first, broad-scale study, and then use the finer-scale work as points of comparison.

The plot-based findings supported our hypotheses that whitebark pine seedling density and proportion are positively associated with more recent disturbances of greater severity. However, the sample size for high severity sites was small, limiting our confidence about increased seedling density following severe disturbance. Support for a high-severity disturbance limit on recruitment is suggested by the positive correlation between seedling density and whitebark pine cover in the main canopy; if disturbance were severe enough to reduce main canopy cover, recruitment may diminish due to loss of seed sources. However, in some cases, loss of the main canopy may not preclude abundant regeneration. For example, Meyer et al. [13] found that increasing severity of canopy cover

loss due to a mountain pine beetle attack was positively associated with whitebark pine seedling density. This was in part attributed to the release of basal sprouts and already established seedlings, and, therefore, loss of the main canopy as a seed source would therefore not necessarily affect seedling density immediately post-disturbance. Studies conducted in the Rocky Mountains have similarly found enhanced regeneration in canopy openings created by mountain pine beetle attacks [22].

For both whitebark pine seedling density and proportion, % cover of whitebark in the main canopy was an important predictor. This is consistent with Maloney [34] and Leirfallom et al. [35], who found seed source pressure, as a function of distance to reproductive trees, to be strongly correlated to recruitment. Alternatively, Gelderman et al. [20] found negative associations between recruitment and basal area. This difference from our study may be explained by the contrasting habitats; generally greater tree cover at northern latitudes may have inhibited seedling success in previous studies. In our study at the southern range limit, mean total tree cover in the main stratum was 29%, with 0% as the most common value at the 0.08 ha scale. In the northern Cascades, Amberson et al. [36] also found no relation between recruitment and stand basal area. They suggest landscape level availability of trees is more important, because whitebark pine's primary seed disperser, the Clark's nutcracker, can travel relatively far [37]. That hypothesis is consistent with our finding that small increases in whitebark pine cover in the southern Sierra Nevada do make a difference. Small increases are in a sense a proxy for landscape level presence of whitebark in this setting.

Disturbance that created relatively limited openings in the main canopy appeared to have the greatest positive effect for recruitment. Whitebark pine seedling survivorship has been shown to be enhanced by surrounding vegetation cover, which affords protection from sky exposure, frost, and damaging sunlight [38,39]. However, our finding that only increased cover of whitebark pine—and not other trees, shrubs, or herbs—enhanced whitebark recruitment suggests the relationship in this setting is driven more by seed availability, rather than through the mechanism of biophysical protection. Indeed, Gelderman et al. [20] found that seedling occurrence was most common on bare mineral soil. These complex relationships highlight the importance of defining disturbance type and severity; our categories examined multiple disturbance types, with a range of effects in both soil disturbance and canopy cover loss. Further work through experimentation that controls disturbance effects to soil vs canopy, as well as time since disturbance, would help in characterizing the exact environmental cues to which recruitment, and, by association, dispersal agents and subsequent retrieval patterns by granivores are responding. The focus of our study was on the stand-level attributes associated with successful recruitment, recognizing that the mechanisms at play are both autecological and synecological in nature [40].

Our findings also demonstrate that seedling density vary by range position, with the greatest numbers occurring at mid latitudes in the Sierra Nevada, and at mid elevations. Such a pattern of regeneration is typical of at-risk species: reproduction is most successful in a portion of the species' range, typically at the range core—a phenomenon known as habitat attenuation [41].

The importance of disturbance varied with range position. Although disturbance increased recruitment at all elevations, disturbance appeared especially important at the lower range limit. Here, disturbance provided the opportunity for recruitment to occur at all, whereas at higher elevations, recruitment was found across all disturbance categories. One mechanism through which disturbance might be relatively more important at the lower range edge is through preference of Clark's nutcrackers for open, patterned habitat, potentially leading them to preferentially deposit caches in stands recently disturbed by fire or tree mortality [42]. Alternatively, the disturbance-range position interaction may have been mediated though effects of climate, which we know to be correlated to range position as we measured it. Again, further work to isolate the separate effects of range position and climate will help answer these questions.

Our second study at recreational sites supports the assertion that the effects of disturbance were dependent on range position. Whitebark pine seedling densities were greatest in severely disturbed areas deeper within the core range, where seed sources were more abundant. At the range edge,

however, very little whitebark pine recruitment occurred in highly disturbed areas. Proportionately, at the range edge, whitebark recruitment was most successful at low levels of disturbance, whereas other species, primarily lodgepole pine, invaded severely impacted sites. These results provide additional evidence for the apparent limit to increasing recruitment at very high levels of disturbance, as suggested by the plot-based study. The complete absence of whitebark pine in the disturbed ski runs of June Mt. is also consistent with this finding. Again, nutcrackers may preferentially select patterned settings and reference landmarks to cache seeds to allow for easier retrieval. This may in part explain why very high levels of disturbance actually limit whitebark pine recruitment.

The June Mt. case study served as a practical example of the interplay between disturbance intensity and range position. Here, whitebark pine recruitment was enhanced at its lower elevation limits by the mountain pine beetle attack, but was suppressed at the lower range edge when the overstory canopy was entirely absent, as in the open ski runs. While the mountain pine beetle attack did not appear to enhance absolute number of whitebark pine seedlings, the proportional benefits to whitebark recruitment were evident: seedlings already present in the understory demonstrated a strong growth release after the attack, whereas lodgepole did not exhibit advance regeneration (i.e., older seedlings present prior to disturbance). Thus, while our plot-based and recreational site studies indicate that sheer numbers of seedlings were enhanced by low level disturbance, the June Mt. case study shows that advance regeneration and subsequent growth release induced by the attack may be equally, if not more, important. In either case, the beetle attack disturbance, which allowed for maintenance of both younger and older seedlings, appears to have created a stand composition, which may shift the lodgepole-whitebark pine ecotone to a lower elevation, if the growth of the whitebark seedlings equals or outpaces that of lodgepole pine.

Our studies suggest that a successional pathway in which whitebark pine gains dominance following disturbance, and loses dominance to other conifers in the absence of disturbance does exist in the Sierra Nevada, as has been observed elsewhere [43]. This was evident from low levels of whitebark recruitment, in absolute and proportional terms, in sites that have experienced less disturbance. However, it is possible that the high levels of recruiting seedlings following disturbance may not survive to join the main canopy and reproduce. Therefore, long-term monitoring to demonstrate the change over time will be required. Our findings do show that if monitoring is to detect that pathway, it must be carefully stratified by range position.

Other studies have documented the contextual importance of succession: Clason et al. [43] found successional replacement of whitebark by subalpine fir (*Abies lasiocarpa* (Hook.) Nutt.) and mountain hemlock in British Columbia, with variation between xeric and submesic stands. Similarly, Campbell and Antos [44] found that whitebark pine recruitment is abundant post-disturbance, but is limited by the presence of other species, especially lodgepole pine in the Canadian Rockies. Amberson et al. [36] found no evidence of successional replacement of whitebark pine following disturbance in the Cascades. However, the limited sample size (despite long-term strength) may have resulted in failure to detect patterns that might vary by spatial context. As a case in point, we found evidence of disturbance effects on seedling composition only through separate consideration of the range edge and core, with disturbance important to composition at the range edge, but not at the core.

Conserving components across a species' range—even at the edge—is important to maintain genotypic and phenotypic diversity to enable adaptation as environmental conditions change [19,45,46]. In the case of whitebark pine, such maintenance is especially important for resistance to white pine blister rust [47]. The low elevation and low latitude edges of populations are thought to have bad prospects in a warming climate. In many cases, populations in edge localities, or refugia, can potentially be wiped out by disturbance [48,49]. However, for disturbance-dependent species, such as whitebark pine, our results demonstrate that opportunity exists for maintenance at the range edge through disturbance effects. This is in contrast to other studies that have shown increasing disturbance frequency and severity reduce the time needed to achieve equilibrium of a novel species composition reflecting new climate regimes [50,51]; for disturbance-dependent species, this may not be the case. Our findings

also provide perspective for studies that have projected strong losses of whitebark pine habitat over the upcoming decades, using climate envelope habitat availability based methods [52]. Potentially, an altered disturbance regime will favor an early seral species like whitebark, thereby dampening the negative effects of a changing climate.

## 5. Conclusions

Whitebark pine recruitment is most common at the range core of the southern Sierra Nevada, at upper to mid elevations, and mid latitudes, but appears to be maintained at lower elevations through disturbances that create canopy openings, including mountain pine beetle attack and recreational use. We emphasize that the beneficial effects of disturbance will most likely be realized when disturbances are of limited size and severity, and of variable types. Management actions that will allow gradual, rather than abrupt change are most likely to support continued persistence of current mountain ecosystems [53]. Furthermore, as we have demonstrated, the interactive relationship between disturbance and range positioning is of critical importance: management actions, such as thinning, prescribed burns, or other impacts, can be expected to have differential outcomes depending on location. This is no surprise, but still deserves explicit recognition in restoration strategies. As a practical example, at lower elevations where other tree species codominate, management that introduces disturbance to encourage whitebark recruitment, but also protects nearby mature cone-bearing trees, may be most effective to maintain whitebark in those areas [30]. Our finding that whitebark continues to be the most successful conifer in the recruiting layer where it is already dominant or codominant in the overstory indicates some apparent stability in stand structure, suggesting that careful management strategies that provide for continued viable seed production and dispersal may be successful for long-term protection.

**Author Contributions:** M.R.S. conceived the research, supervised the field campaign for the first two studies, conducted the analysis, and led the writing of this manuscript; M.M. conducted the field work for the case study and contributed to the writing of this manuscript; T.K. contributed to the methodology, analysis, editing, and review; C.M.R. led the resource administration that made this project possible, and contributed to the analysis and review of this manuscript.

**Funding:** This research received no external funding.

**Acknowledgments:** We thank field assistants Paul Slaton, Sue Weis, Paul Sattherthwaite, and Sophia Winitsky. We gratefully acknowledge the Inyo National Forest for logistical support of this research and analysis.

**Conflicts of Interest:** The authors declare no conflict of interest.

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
