# Peer review of "Whitebark Pine Recruitment in Sierra Nevada Driven by Range Position and Disturbance History"

_forests, doi:10.3390/f10050455_

Round 1

Reviewer 1 Report

General comments

This manuscript describes the results of three observational studies examining the abundance of whitebark pine seedings.  An interesting facet of this work is the degree it attempts to put results in the  context of this species range.  The downside of the work is that the studies are not particularly rigorous in their design, although I believe the authors do make the case that taken together the evidence is sronger.  I can certainly accept that logic, but there are other issues that are harder to gloss over.  For example, in the first study the period of data collection was quite long and perhaps long enough that climatic responses would be present.  I was left wondering about some of the methods related to aging, particularly of herbs and shrubs.  These seemed potential quite subjective.  The area of most concern deals with the statistical analyses and their interpretation.  Most worrying was the fact that climate variables seemed to be correlated to some of the locational indices, but the authors decided to select one over the other based on their hypotheses/interests.  This does not seem objective and clearly one cannot possibly conclude anything about cause and effect.  At the very least the authors need to be less assertive about their results.  Not only is this correlation of potential explanatory variables an issue, but a close examination of the scatter plots indicates many of the relationships are quite weak.  This is not unexpected in this sort of study, but the authors need to offer their results as possible relationships and possible insights rather than firm conclusions.    

Specific comments (line)

32 Do you mean the white bark pine zone or forest? I find it hard to believe the snow only falls on the trees and does not touch the ground as this seems to imply.

65 Do these refer to positions in the range? That is not clear as written. Also what is the leading edge?  This implies the range is always moving in one direction.  Doesn’t that depend on the way climate is changing?

69 to be fair to the other authors one has to recognize that when multiple threats are listed, they could all be occurring somewhere in the range and the statement “throughout the range” would still be true.  I take the current authors point that it is important to understand and appreciate different responses in different locations or parts of the range.  My point is that this view need not supplant the previous view, it just adds important details. Perhaps a more positive set of points could be made here without attacking past work?

86 Perhaps “the species’ total distribution” would be a less confusing way to state this?  By writing total species distribution it implies there are other species involved.  Also doesn’t the distribution belong to whitebark pine?  So would that not imply possessive?

119 decades could cover anything from 10-100 years. Can the authors be more specific?

125 How was the canopy viewed from above?

129 It is not clear how one could age the shrubs even approximately.  How was this done?

133 How does one assess recruitment of herbs?  Were the authors looking for species that indicated disturbance?

174 I was a bit confused by this section. It seems to state that climate is important, but not used, but correlated with elevation, but is important. 

511 whitebark pine?

518 is the species name really whitebark? Or is that jargon?

524 Why are variable types of disturbance favorable to whitebark pine?  I was not sure that point was uncovered by the analysis.

526  How is possible to remove mountain ecosystems?  Do the authors mean the current ecosystem/community?  This just seems overly general as a statement if not conceptually dubious.

Author Response

Please see the two attached files: Cover Letter and Point-by-Point response to each reviewer.

Reviewer 2 Report

A very interesting study on recruitment of whitebark pine.  The paper is nicely organized and clearly written.  You do an excellent job of linking information on species ecology and dispersal with discussion of results, which makes this paper particularly informative and useful to management.  I do wonder if it might be useful to include a few paragraphs on seed release and seed dispersal, particularly wrt Clarks nutcracker, in the introduction.

A couple of minor English corrections:  line 448 - remove the second "that"; line 480 - remove "so" and just say "if not more, important".

Author Response

Please see two attached files: Cover Letter and Point-by-Point responses to each reviewer.
